# Study of the Reinforcement Effect in (0.5–x)TeO_2_–0.2WO_3_–0.1Bi_2_O_3_–0.1MoO_3_–0.1SiO_2_–xCNDs Glasses Doped with Carbon Nanodiamonds

**DOI:** 10.3390/nano12193310

**Published:** 2022-09-23

**Authors:** Artem L. Kozlovskiy, Indira Tleulessova, Daryn B. Borgekov, Vladimir V. Uglov, Viktor M. Anishchik, Maxim V. Zdorovets, Dmitriy I. Shlimas

**Affiliations:** 1Laboratory of Solid State Physics, The Institute of Nuclear Physics, Almaty 050032, Kazakhstan; 2Engineering Profile Laboratory, L.N. Gumilyov Eurasian National University, Nur-Sultan 010008, Kazakhstan; 3Department of Solid State Physics, Belarusian State University, 220030 Minsk, Belarus

**Keywords:** reinforcement, telluride glasses, protective shielding materials, optical transmission, hardening

## Abstract

The purpose of this study is to examine the influence of carbon nanodiamonds on the reinforcement and hardening of telluride glasses, as well as to establish the dependence of the strengthening properties and optical characteristics of glasses on CND concentration. According to X-ray diffraction data, the synthesized glasses have an amorphous structure despite the addition of CNDs, and at high concentrations of CNDs, reflections characteristic of small crystalline particles of carbon nanodiamonds are observed. An analysis of the strength properties of glasses depending on the concentration of the CND dopant showed that an increase in the CND concentration to 0.10–0.15 mol. leads to an increase in hardness by 33–50% in comparison with undoped samples. The studies carried out to determine the resistance to external influences found that doping leads to an increase in the resistance of strength characteristics against destruction and embrittlement, and in the case of high concentrations, the change in strength properties is minimal, which indicates a high ceramic stability degree. The study of the radiation resistance of synthesized glasses found that the addition of CNDs leads to an increase in resistance to radiation damage when irradiated with gamma rays, while also maintaining resistance to high radiation doses. The study of the shielding characteristics found that the addition of CNDs is most effective in shielding gamma rays with energies of 130–660 MeV.

## 1. Introduction

One of the promising directions in the development of modern materials science is research in the field of protective shielding materials and their practical application. The interest in this area is due to the possibility of increasing the use of ionizing radiation sources in various industries related to everyday life, as well as the development of space technologies and space exploration [1,2,3].

It is known that the use of ionizing radiation sources is accompanied by the impact of ionizing radiation not only on the object of irradiation (as, for example, in medicine), but also on the environment, which, with prolonged exposure, can cause negative consequences [4,5]. In the case of cosmic radiation, the greatest danger is posed by gamma radiation and neutron fluxes, which, due to their nature, have a high penetrating power and can initiate various sporadic effects in microelectronic devices. Also, due to the compactness and the desire to miniaturize microelectronic devices, meteor showers can be a great danger to them, as well as protective cases, the result of which is mechanical damage upon impact [6,7,8].

In turn, the use of radiation-resistant protective materials introduces certain limitations, mostly related to the overall dimensions and the desire to increase payload volumes [9,10]. At the same time, it is worthy to note that it is quite difficult to predict exactly what effect ionizing radiation will have on microelectronic devices, as well as when and under what circumstances this will happen due to the probabilistic nature of the interaction of radiation with the material, as well as the chaotic behavior of ionizing radiation in the material [11,12,13].

One of the working methods of protection against the negative effects of ionizing radiation, the development of which is directed by a fairly large number of studies, is the use of amorphous or amorphous-like ceramics and glasses as shielding materials [14,15,16,17]. Interest in these types of materials is due to their combination of properties that provide sufficiently high performance in the field of shielding ionizing radiation and reducing its effects, comparable in efficiency to traditional protective materials based on lead and concrete [18,19,20,21,22]. So, for example, in recent years, much attention has been paid to the development of new types of glasses based on oxide compounds such as zinc borate glasses [23,24], Na_2_O-BaCl_2_-B_2_O_3_ glasses [25,26], and telluride glasses with various dopants, the choice of which is due to the possibility of increasing the screening efficiency [27,28,29,30]. Great interest in this area is due to the possibility of increasing the shielding characteristics of materials, as the development of new areas of research on the creation of protective shielding materials is necessary for the widespread use of ionizing radiation sources.

In most works known to date, the main emphasis has been on increasing the efficiency of screening ionizing radiation, which is provided by varying the elemental composition of the glass components, as well as their content. However, such a factor as resistance to external influences, whether mechanical damage or thermal effects, is practically not considered; however, for several reasons, these effects are very important to consider when calculating the performance characteristics of the proposed protective materials, as well as assessing the prospects for their use.

The purpose of this study is to investigate the influence of carbon nanodiamonds on the reinforcement and hardening of telluride glasses, as well as to establish the dependence of the strengthening properties and the optical characteristics of glasses on CND concentration. The choice of carbon nanodiamond as a dopant is due to the possibility of increasing the strength properties of glasses against mechanical damage due to the reinforcement effect associated with the material strengthening and the creation of additional centers preventing the propagation of cracks and cleavages under external influences. The novelty of the work consists in its consideration of the possibility of telluride glass modification using carbon nanostructures to increase resistance to external influences. The paper also considers the results of using these glasses as shielding materials for protection against gamma radiation and the effect of CND doping on the strength properties and radiation damage resistance.

## 2. Experimental Section

Telluride glasses of the following composition (0.5–x)TeO_2_–0.2WO_3_–0.1Bi_2_O_3_–0.1MoO_3_–0.1SiO_2_–xCNDs were chosen as objects for research. The choice of carbon nanodiamonds as a dopant was due to the possibility of increasing the strength properties of the glasses against mechanical damage due to the reinforcement effect. The addition of CNDs was carried out at the stage of sample mixing during the mechanochemical grinding of the initial components to obtain a uniform distribution of elements. After stirring, the resulting mixtures were placed in a muffle furnace and heated to a temperature of 900 °C, at which the mixture straightens and vitrifies when the samples are quickly removed from the muffle furnace. Samples were annealed in alundum crucibles capable of withstanding heating temperatures up to 1800 °C. After extraction, the samples were placed in a box with an inert atmosphere in order to avoid oxidation processes during quenching and cooling. After extraction and subsequent cooling, the samples had sufficient strength against mechanical stress during further processing and polishing. The presence of stress in the structure did not lead to their rapid cracking during processing nor during further exposure.

To visualize CNDs in the initial state, scanning electron microscopy was used, performed on a Jeol 7500F microscope (Jeol, Tokyo, Japan) at an accelerating voltage of 5 kV, and images were taken in the SEI mode.

Morphological features and determination of the distribution of CNDs in amorphous glass were performed using high-resolution transmission electron microscopy, implemented on a Jeol JEM 1400 Plus transmission electron microscope (Jeol, Tokyo, Japan). Images were taken at an accelerating voltage of 120 kV, exposure time 4 s. To determine the distribution of elements in the compositions of the glasses, the method of energy dispersive analysis was applied.

For convenience of notation, the samples were further numbered next using the following abbreviations indicated in Table 1.

Doping of CNDs was carried out by replacing the TeO_2_ component with stirring and subsequent sintering. The choice of dopant concentration was made based on a priori data from a study of the effects of reinforcement and hardening of materials with the addition of carbon nanostructures.

Determination of the amorphous nature of the synthesized glasses and its change as a result of CND doping was carried out using the X-ray diffraction method implemented using a D8 Advance Eco X-ray diffractometer (Bruker, Berlin, Germany).

Determination of the optical properties of the glasses depending on the concentration of the CND dopant in the glass composition was carried out by measuring the optical spectra in the UV-Vis range (350–1000 nm) using a UV-Vis spectrometer (Jena Specord-250) using an integral sphere. The band gap was calculated using the Tauc plot analysis method and the construction of approximating straight lines. To measure the optical properties, particularly the linear refractive index, the samples were polished. The determination of the refractive index was carried out at a wavelength of 587.6 nm.

The calculations of optical characteristics such as linear refractive index (*n^optical^*), optical transmission (*T^optical^*), refraction loss (*R^loss^*), and static dielectric constant were carried out using Equations (1)–(4) [31,32]:(1)$$\frac{\left[{\left(n^{optical}\right)}^{2}-1\right]}{\left[{\left(n^{optical}\right)}^{2}+2\right]}=1-\sqrt{\frac{E_{g}}{20}}$$
where *E_g_* is the band gap.
(2)$$T^{optical}=\frac{2\left(n^{optical}\right)}{{\left(n^{optical}\right)}^{2}+1}$$
(3)$$R^{loss}={\left(\frac{\left(n^{optical}\right)-1}{\left(n^{optical}\right)+1}\right)}^{2}$$
(4)$$\varepsilon^{static}={\left(n^{optical}\right)}^{2}$$

Testing of glass samples to determine the strength characteristics and determine the influence of the reinforcement effect associated with the addition of CNDs was carried out using the following methods. The initial hardness and hardening as a result of doping were measured using the microindentation method using a LECO LM 700 microhardness tester. The crack resistance was measured by the method of single compression of the samples at a constant compression rate of 0.1 mm/min. The use of this method made it possible to establish the resistance of the material to mechanical stress, as well as determine the amount of pressure that the glass could withstand before the formation of cracks and cleavages. The resistance of the glasses to the aging effect was determined using a simulation of the environment and its impact over a long period of time, leading to aging and brittleness of the material. The aging effect was simulated by exposing the glass to water vapor under pressure for 30 h, simulating the natural aging effect comparable to 3–5 years of environmental exposure under natural operating conditions.

The efficiency of shielding characteristics influenced by CND doping was studied using the method of recording gamma radiation with fixed energies (130 keV, 660 keV, 1270 keV), as well as comparing the change in the intensity of gamma radiation before and after shielding using the obtained glasses with a thickness of 5 mm. Sources of Co^57^ with a gamma-ray energy of 130 keV, Cs^137^ with a gamma-ray energy of 660 keV, and Na^22^ with a gamma-ray energy of 1270 keV were used as gamma-ray sources with fixed energies. The use of these sources made it possible to evaluate the shielding efficiency of protective materials in the energy ranges of gamma rays, which are characterized by different types of gamma rays interactions with the material, depending on the energy: photoelectric effect (Eγ ≤ 0.1 MeV), Compton effect (Eγ ≥ 0.1 MeV), and the formation of electron-positron pairs (Eγ ≥ 1 MeV). Shielding data were obtained from a series of experiments, which consisted in determining the change in the gamma ray set intensity when it passed through a protective glass, followed by comparison with a similar value recorded without a shield. To generate gamma rays, standard sources were used located at a distance of 10 cm from the protective glass, behind which a NaI detector was placed, which recorded the gamma radiation intensity.

The shielding efficiency (*RFE*) was determined using Formula (5):(5)$$RFE=\left(1-\frac{I}{I_{0}}\right)\times 100\%$$
where *I* and *I*_0_ are the intensities of gamma radiation before and after passing through the protective shield and were recorded by the NaI detector.

The linear absorption coefficient (LAC) was determined using Formula (6):(6)µ=lnI0Id
where *d* is the glass thickness.

The radiation damage resistance of the samples was determined by irradiating the samples at the ELV-4 accelerator with a 1.3 MeV gamma-ray flux with preset radiation doses of 100, 200, and 500 keV.

## 3. Results and Discussion

### 3.1. Study of the Structural and Optical Properties of Synthesized Glasses

Figure 1 shows the X-ray diffraction results of the studied glasses depending on their concentration of CNDs. The general view of the diffraction pattern for the TWBMSND-0 sample is characteristic of an amorphous substance that does not have obvious diffraction reflections characterizing structural compounds of a crystalline nature. When CNDs are added to the composition of glasses at low concentrations of 0.01–0.05 mol, no significant changes in the amorphous nature of diffraction patterns are observed. However, in the case of an increase in the concentration of CNDs to 0.10 mol and above, the formation of a characteristic broadening of the halo in the region of 2θ = 40–45° is observed, which, when described by a certain set of Gaussian lines, can determine the presence of strongly broadened low-intensity reflections characteristic of CNDs, similar to the results of [33]. The presence of such broadened reflections may be due to an increase in the concentration of CNDs in the composition of glasses, which leads to their identification by X-ray diffraction and indicates that the incorporation of CNDs occurs with the preservation of their crystalline form.

Figure 2 shows the results of changes in the optical transmission of glasses with different concentrations of the CND dopant, reflecting the effect of doping on the change in the optical properties of glasses, as well as the shift of the fundamental absorption edge. As can be seen from the data presented, in the case of TWBMSND-0 glass, the optical spectrum is characterized by a fairly good transmission value (more than 75%) and also by a fundamental absorption edge in the region of 370–390 nm.

According to the data obtained, when CNDs are added to the composition of glasses, a sharp change in the transmittance of the glass is observed in the region characteristic of visible light and the near IR range, and in the case of CND dopant concentrations of 0.10–0.15 mol, a threefold decrease in the transmission value is observed compared to the initial composition of the glass. Such a decrease may be due to the formation of additional absorption centers with an increase in the concentration of CNDs, as well as the formation of additional interfacial boundaries that prevent the passage of light in the material. A decrease in the transmission value indicates an increase in the absorbing ability of glasses with an increase in the concentration of the CND dopant, which can also have a significant effect on the optical properties of glass.

The results of estimating the fundamental absorption edge and its shift depending on the concentration of the CND dopant in the composition of glasses (presented in Figure 3) indicate that with an increase in the concentration of CNDs, the band gap decreases, as well as the shift of the fundamental absorption edge. At the same time, the most pronounced changes in the band gap are observed at a CND concentration of more than 0.10 mol, for which the decrease in the band gap is more than 30–35%. As can be seen from the data presented, samples with a high dopant content have low optical transmission, which makes it difficult to use these glasses as optically transparent shielding glasses. However, using them as shielding materials in places where direct visual observation or high optical transparency is not needed, these glasses can be used as protective shielding materials.

Table 2 presents the results of calculating the optical characteristics of the studied glasses depending on the concentration of the CND dopant. For calculations, the results of changes in the optical spectra, the construction of Tauc plots, and calculation Formulas (1)–(4) were used. The general trends of the changes in optical quantities characterize the changes in the optical properties, reflecting, and absorbing ability of glasses with a change in the CND dopant concentration. As can be seen from the data presented, a decrease in the band gap leads to an increase in the refractive index, as well as an increase in reflection losses and an increase in the static permittivity, which indicates an increase in the absorbing ability of glasses. The increase in the refractive index, in turn, is due to the formation of additional absorbing centers in the glass structure, which can reduce the intensity of light transmission and increase the absorption capacity.

### 3.2. Study of the Strength Properties of Glasses and Their Resistance to External Influences, including Radiation Resistance to Irradiation with Gamma Rays

Figure 4 shows the results of changes in the strength characteristics of glasses obtained using the indentation and single compression methods. The results of these changes reflect the resistance of glass materials to external influences, as well as changes in strength indicators depending on the dopant concentration.

As can be seen from the data presented, in the case of the undoped glasses, the hardness value was more than 300 HV, and the maximum pressure that the glass could withstand at a single compression was more than 20 N. These indicators demonstrate rather average strength characteristics of glass, which may limit their scope under conditions of high mechanical loads. Doping with CNDs at a concentration of 0.01–0.05 mol leads to an increase in hardness by 3–10% and crack resistance by 6–22%, which indicates that the addition of CNDs creates a positive effect by strengthening the glasses and increasing their resistance to mechanical pressure. Increasing the concentration of CNDs to 0.10 mol leads to an increase in glass hardness by more than 30% and resistance to cracking and microcracking by more than 50%. This increase is due to the reinforcement effect, which is most pronounced at a dopant concentration of 0.15 mol. In the case of a CND concentration of 0.15 mol, the strength of the glasses increases more than 50%, while the crack resistance exceeds the initial value by more than 2 times. The increase in the strength and hardness of glass with an increase in the CND concentration can be explained by the effect of the appearance of additional interfacial boundaries between CNDs distributed in the amorphous glass volume, which leads to an increase in density and resistance to external influences.

Figure 5a shows the results of morphological studies of CNDs in the initial state, obtained using the method of scanning electron microscopy. According to the data presented, the shape and size of the initial CNDs are almost isotropic, with an insignificant number of larger particles present. The image was made using a scanning electron microscope on a substrate of conductive carbon tape.

Figure 5b shows TEM images of glasses doped with CNDs, in which it is clearly seen that CNDs are embedded in the glass and there are interfacial boundaries between the amorphous volume and crystalline CNDs. It is also worth noting that the presence of interphase boundaries causes a decrease in throughput and an increase in the refractive index. Figure 5c shows a TEM image of a side cleavage of glass, which shows the absence of large agglomerates of CNDs and a close-to-isotropic distribution of CNDs throughout the glass volume.

Figure 5d shows the results of mapping the area containing an inclusion in the form of single CNDs, the main purpose of which is to reflect the absence of oxidative processes of CNDs during thermal sintering, as well as to establish the isotropy of the distribution of other elements in the glass composition. As can be seen from the presented mapping data, no oxidation of CNDs was observed after thermal annealing, which is confirmed by the absence of oxygen on these maps. At the same time, the remaining elements have an almost isotropic distribution over the structure, which indicates the homogeneity of the glass composition.

One of the important characteristics for protective materials is their resistance to long-term mechanical stress during operation, as well as the preservation of strength characteristics for a long time under various external influences. The resistance of glass to long-term mechanical stress, including mechanical pressure, impact, or friction arising during their operation as protective materials is a significant parameter. Thus, one of the requirements for materials used as shielding materials is to maintain strength and crack resistance for a long time under external mechanical pressures, friction, and impacts. At the same time, maintaining resistance to external influences including exposure to the atmosphere, temperature, pressure, impact, and corrosion is one of the important parameters for long-term operation.

Figure 6 shows the results of the experiments carried out to determine the stability of the strength characteristics of the glasses as a result of the simulation of natural aging processes. These results make it possible to evaluate how resistant the obtained glasses are to external influences when exposed to the external environment, as well as the effect of CND doping on the retention of strength characteristics. Determination of the stability degree was carried out by exposing the glass to water vapor under pressure for a certain time, during which the hardness and crack resistance were measured at equal time intervals.

An analysis of the changes in hardness values depending on the testing time showed that in the case of undoped samples, a decrease in hardness occurs after 15 h of exposure to model environmental conditions, which indicates a deterioration in the strength characteristics of the material and a decrease in its resistance to external influences. At the same time, the maximum decrease in the hardness of the samples was more than 25% of the initial value after 30 h of testing, which indicates a strong destruction of the strength characteristics of the samples without a dopant, as well as a high degradation rate in the simulation of aging processes. A high degree of reduction in strength characteristics indicates a weak resistance of undoped glasses to processes comparable to the impact of the external environment during aging.

As can be seen from the data presented, in the case of samples containing CND dopant with a concentration of 0.01 and 0.05 mol, a decrease in hardness as a result of external influences was also observed after 15 h of testing, however, the decrease was much less than in the case of undoped samples. Such a decrease in the softening degree and destruction during aging simulation for doped samples can be explained by the reinforcement effect, which affects not only the increase in hardness and crack resistance, but also reduces the rate of degradation to softening and embrittlement. The addition of CNDs to the composition of glasses leads to a decrease in strength degradation by 10 and 15% at dopant concentrations of 0.01 and 0.05 mol.

In the case of an increase in the CND dopant to 0.10 mol, a fivefold decrease in the change in hardness indices was observed after 30 h of testing, and in the case of a dopant concentration of 0.15 mol, the value of ∆H was less than 1.5%, which indicates the high resistance of the glasses to destruction and aging processes.

Analyzing the trends in hardness changes as a result of the aging simulation, a number of conclusions can be drawn. Firstly, the doping of CNDs in low concentrations leads to a decrease in the rate and degree of destruction of the surface layer and its strength characteristics, which is due to the reinforcement effect and the increase in hardness. Secondly, an increase in the CND dopant concentration to 0.10–0.15 mol leads to an increase in strength and resistance to external influences, and the change in hardness indices after a series of tests was within the permissible error limits.

Thus, based on the data obtained, it can be concluded that the addition of CNDs to the composition of glasses not only has a positive effect on their strength characteristics, but also increases resistance to external influences, which will allow them to be used in more severe operating conditions.

To determine the resistance of the synthesized glasses to radiation damage caused by irradiation with gamma rays, several experiments were carried out including the irradiation of glass samples with gamma rays with an energy of 1.3 MeV at specified doses of 100, 200, and 500 kGy, followed by the measurement of strength characteristics. Figure 7 shows the results of changes in the hardness of the studied glasses irradiated with gamma rays at different doses.

As can be seen from the data presented, CND doping leads to an increase in the radiation damage resistance of the synthesized glasses, which causes a decrease in strength characteristics, which is observed for undoped glasses. At the same time, the greatest decrease is observed when the radiation dose exceeds 200 kGy, which indicates the softening of materials as a result of the radiation damage accumulation. At the same time, for samples with a dopant concentration of more than 0.05 mol, a decrease in hardness for irradiated samples is observed after irradiation with a dose of 500 kGy, while the decrease is no more than 2–3%, which is within the measurement error. Such a small change indicates a rather high radiation damage resistance of glasses doped with CNDs upon irradiation with gamma rays.

### 3.3. Study of the Gamma Radiation Shielding Efficiency of Synthesized Glasses Depending on the CND Dopant Concentration

Figure 8 shows the results of the efficiency of shielding and the reduction of gamma ray intensity with different energies as they pass through the synthesized glasses.

The general view of the presented shielding efficiencies can be divided into two stages. The first stage is typical for shielding gamma rays with an energy of 130–660 keV, which is characterized by rather high efficiency rates of more than 60% in the case of undoped glasses and close to 80–90% in the case of doped glasses. The second stage is typical for gamma rays with an energy of 1270 keV, for which the shielding efficiency in the case of undoped glasses does not exceed 20%, while for doped glasses, the efficiency is close to half the attenuation of the gamma radiation intensity, which is a very good indicator for these glasses in comparison with similar materials.

Such a big difference in shielding efficiency consists in the mechanisms of gamma rays’ interaction with the glass material, as well as the processes that occur as a result of such interactions. In the case of low-energy gamma quanta (0.1–0.6 MeV), an increase in absorption for doped glasses is due to the dominance of the absorption of gamma quanta during the photoelectric effect, due to an increase in the cross-section of the photoelectric effect on carbon. In this case, doping with CNDs leads to an increase in the absorption cross-section, thereby reducing the gamma radiation intensity due to absorption. In this case, a change in the electron density and, consequently, an increase in the absorbing capacity (absorption) leads to most of the gamma quanta being absorbed during interaction with the electrons of the shells of atoms.

In the case of gamma rays with an energy of 660 keV, the main mechanisms of interaction and, consequently, the weakening of the intensity are the processes of incoherent scattering of gamma rays on electrons (the Compton effect). For these processes, the greatest contribution to shielding and absorption is made by the effect of changing the density of the material being doped with CNDs, which is expressed in the formation of strongly broadened reflections on the diffraction patterns (see Figure 1). At the same time, the evaluation of glass density using the standard method of Archimedes showed that the addition of CNDs leads to an increase in glass density (see data in Figure 8b). Comparing the data on changes in the density of the glasses and the increase in the efficiency of shielding gamma rays with an energy of 660 keV depending on the increase in the concentration of the dopant showed a strong correlation between these values, which indicates a positive effect of the influence of the CND dopant on the shielding and absorption of gamma radiation intensity.

When shielding gamma rays with an energy of 1270 keV, the selected glasses showed average results in terms of efficiency, however, for doped glasses, the shielding indicators were close to half the attenuation of the gamma radiation intensity, which are very good results for shielding high-energy gamma rays. In this case, the dominant process in the interaction of gamma rays with the glass material is the formation of electron–positron pairs, which can cause the effect of secondary radiation in the structure. At the same time, the doping of CNDs, which leads to the creation of additional absorption centers, reduces the efficiency of gamma radiation by more than 1.5 times compared to undoped glasses.

Figure 9 shows the results of the linear absorption coefficients, calculated using Formula (6) and reflecting the dopant’s effect on the absorbing and shielding abilities of the synthesized glasses. As can be seen from the data presented, the addition of the CND dopant leads to a significant increase in the efficiency of linear attenuation of gamma rays with energies of 130–660 keV, while during shielding of gamma rays with an energy of 1270 keV, an increase in the CND dopant in comparison with samples depending on the dopant concentration does not lead to a significant increase in shielding, while this value has a significant increase in comparison with undoped glasses.

To compare the shielding efficiency of synthesized glasses with other types of shielding materials that are proposed as alternative materials for protection against the negative effects of ionizing radiation, several research papers were selected to study the shielding efficiency of various materials. Comparative analysis was carried out by comparing the half-value layer (HVL) for gamma rays with an energy of 660 keV; the results of the comparative analysis are presented in Figure 9b. The following types of glasses were selected for comparison: 60TeO_2_–20PbO-(20–x)ZnO–xBaF_2_ glasses with different concentrations of BaF_2_ dopant [34], (100–x)(60TeO_2_–40PbO)–xB_2_O_3_ glasses with different B_2_O_3_ dopant concentrations [35], TeZnNa–tellurite glass system containing ZnO and Na_2_O [36], TeO_2_–CdO–PbO–B_2_O_3_ glasses [37].

As can be seen from the presented diagram, the addition of the CND dopant leads to a decrease in the HVL value by about 10–40% compared to undoped glasses. At the same time, the synthesized glasses are quite effective in absorption of gamma radiation with an energy of 660 keV in comparison with other types of telluride glasses containing lead oxide, which is also a significant component for creation of shielding materials.

## 4. Conclusions

The paper presents the results of a study of the effect of doping with CNDs at different concentrations on the strength properties of (0.5–x)TeO_2_–0.2WO_3_–0.1Bi_2_O_3_–0.1MoO_3_–0.1SiO_2_–xCNDs glasses. The main purpose of this work was to study the efficiency of the influence of the dopant concentration on increasing the resistance of glasses to external influences due to the appearance of the reinforcement effect. During the studies, it was found that the addition of CNDs led to an increase in strength characteristics, and the maximum increase in hardness and crack resistance was more than 50 and 100% at a dopant concentration of 0.15 mol. An analysis of the results of experiments that simulated natural aging processes showed that an increase in the dopant concentration leads to glass strengthening and an increase in resistance to external influences.

During the studies of gamma radiation shielding at various energy levels, it was found that glasses doped with CNDs at a concentration of 0.1 mol are the most effective in comparison with undoped glasses. At the same time, these glasses are most effective for shielding gamma rays with energies up to 1.0 MeV, which are characterized by interaction processes in the form of the photoelectric effect and the Compton effect.

## Figures and Tables

**Figure 1 nanomaterials-12-03310-f001:**
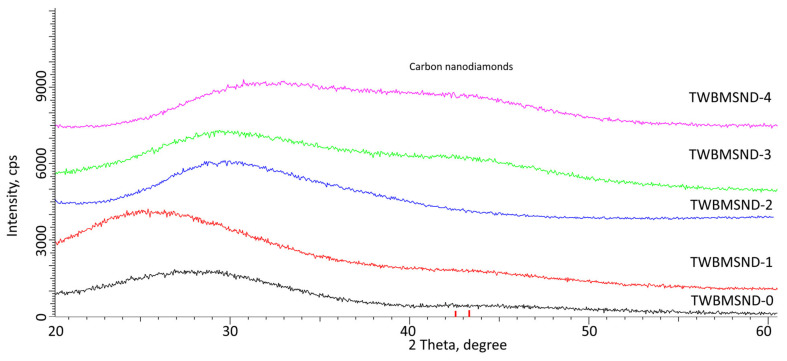
Results of X-ray diffraction of the studied glass compositions depending on the concentration of CNDs (red dashed lines indicate the positions of reflections characteristic of CNDs).

**Figure 2 nanomaterials-12-03310-f002:**
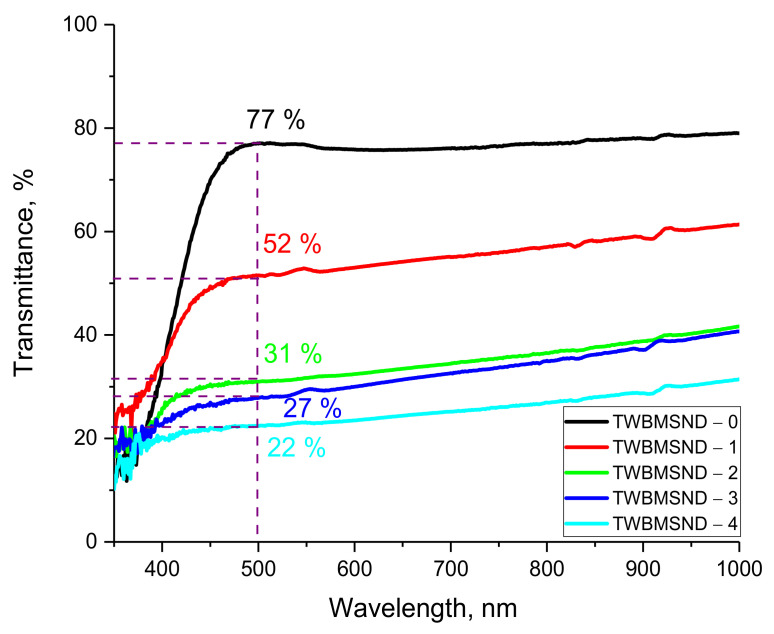
Graph of the optical transmission value depending on the wavelength for glasses with different concentrations of the CND dopant.

**Figure 3 nanomaterials-12-03310-f003:**
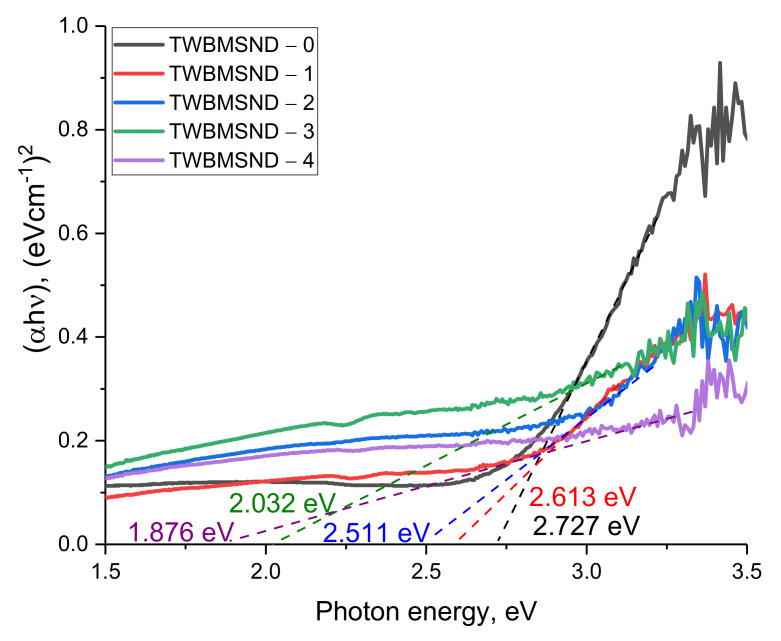
Tauc plot construction results for band gap determination.

**Figure 4 nanomaterials-12-03310-f004:**
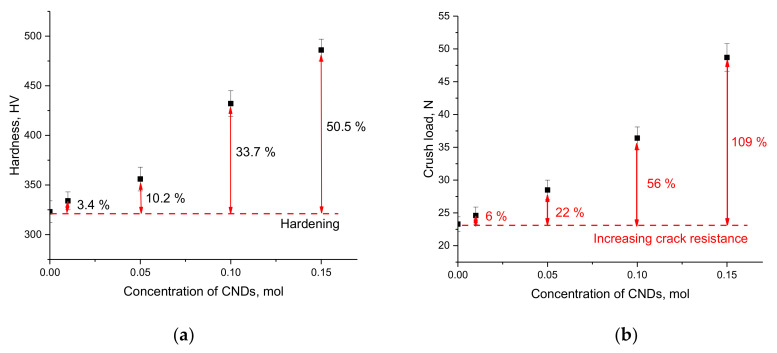
(**a**) Graph of the changes in the glass hardness depending on the CND dopant concentration; (**b**) Graph of the change in the crack resistance value at a single compression depending on the CND dopant concentration.

**Figure 5 nanomaterials-12-03310-f005:**
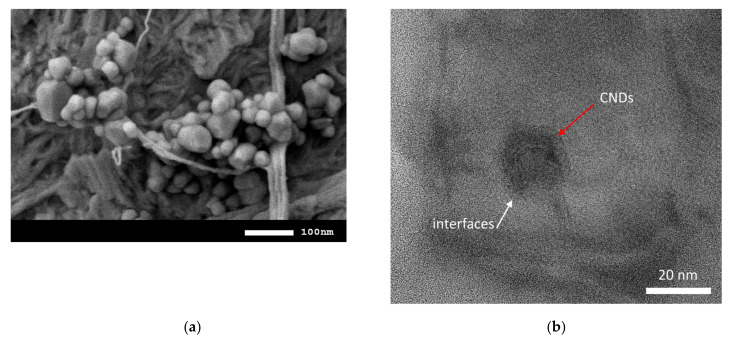
(**a**) SEM image of CNDs in the initial state; (**b**) TEM image of the presence of embedded CNDs in an amorphous glass volume (sample TWBMSND-4); (**c**) TEM image of a side glass cleavage reflective distribution of CNDs in the glass volume; (**d**) Mapping results.

**Figure 6 nanomaterials-12-03310-f006:**
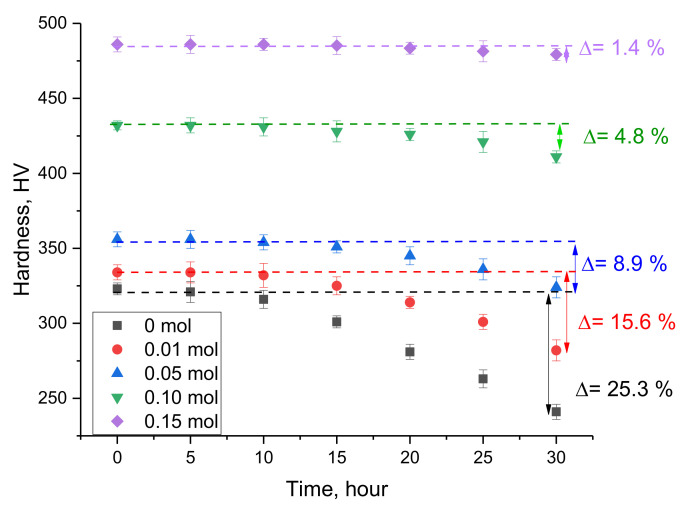
Results of changes in the hardness value during the simulation of aging processes.

**Figure 7 nanomaterials-12-03310-f007:**
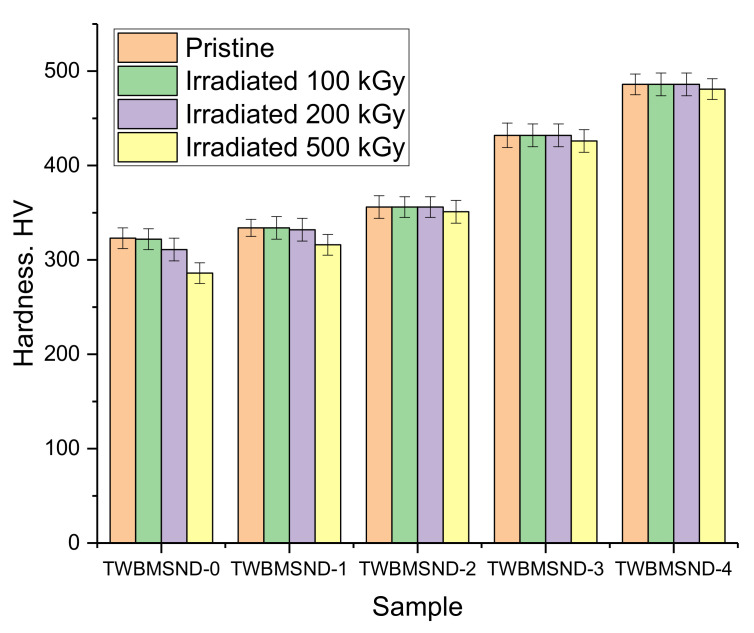
Results of changes in the hardness of samples depending on the irradiation dose.

**Figure 8 nanomaterials-12-03310-f008:**
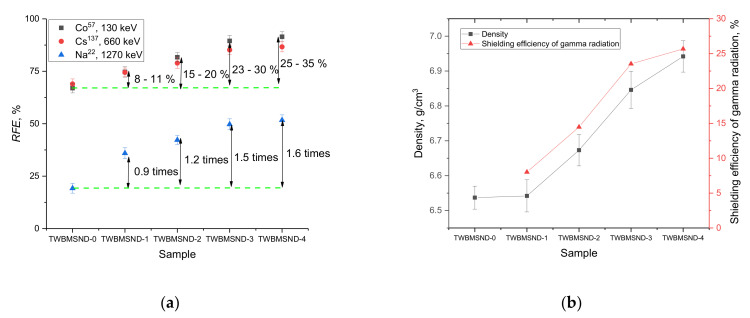
(**a**) Results of evaluation of the efficiency of shielding gamma radiation with different energies; (**b**) Comparison of the change in the density of glasses with the increase in the efficiency of shielding of gamma rays with an energy of 660 keV.

**Figure 9 nanomaterials-12-03310-f009:**
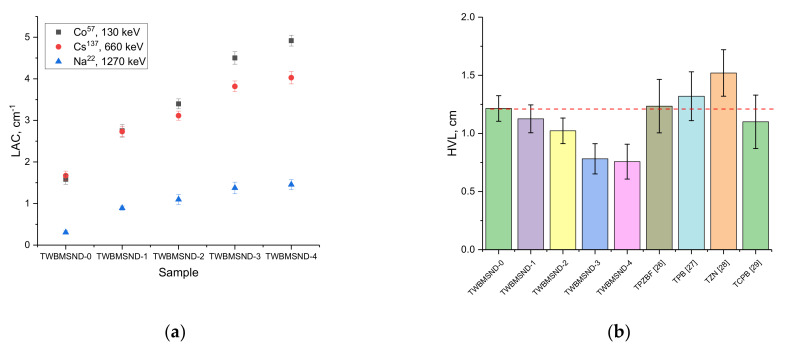
(**a**) Graph of the change in the linear attenuation coefficient for the glasses under study; (**b**) Comparison diagram of HVL values for different types of glass.

**Table 1 nanomaterials-12-03310-t001:** Designation of samples according to the concentration of the components of the glass composition.

Sample	Concentration, mol					
	TeO_2_	WO_3_	Bi_2_O_3_	MoO_3_	SiO_2_	CNDs
TWBMSND-0	0.5	0.2	0.1	0.1	0.1	-
TWBMSND-1	0.49	0.2	0.1	0.1	0.1	0.01
TWBMSND-2	0.45	0.2	0.1	0.1	0.1	0.05
TWBMSND-3	0.4	0.2	0.1	0.1	0.1	0.10
TWBMSND-4	0.35	0.2	0.1	0.1	0.1	0.15

**Table 2 nanomaterials-12-03310-t002:** Data on the optical characteristics of the glasses under study.

Sample	Refractive Index (*n^optical^*)	Reflection Loss (*R_L_*)	Optical Transmission (*T_optical_*)	Static Dielectric Constants (*ε^static^*)
TWBMSND-0	2.474	0.180	0.694	6.12
TWBMSND-1	2.509	0.185	0.687	6.29
TWBMSND-2	2.543	0.189	0.681	6.46
TWBMSND-3	2.722	0.214	0.647	7.41
TWBMSND-4	2.792	0.223	0.635	7.79

## Data Availability

Not applicable.

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
