# Peer review of "Study of the Reinforcement Effect in (0.5–x)TeO2–0.2WO3–0.1Bi2O3–0.1MoO3–0.1SiO2–xCNDs Glasses Doped with Carbon Nanodiamonds"

_nanomaterials, 2022, doi:10.3390/nano12193310_

Round 1
Reviewer 1 Report
|
|||
Author Response
|
Review Report Form 1 |
|
|
1- Avoid the abbreviation in the abstract section.
|
The authors thank the reviewer for this remark, corrections have been made to the text of the article. |
|
2- Enhance the abstract by the output findings
|
The authors thank the reviewer for this remark. Corrections have been made to the text of the article, the abstract has been extended. |
|
3- Focus on the novelty of the work.
|
The authors thank the reviewer for this remark, corrections have been made to the text of the article. The novelty of the work consists in the consideration of the possibility of telluride glasses modification using carbon nanostructures to increase resistance to external influences, and the main results obtained during this study can later be applied to other types of glasses, in particular those containing lead oxide or quartz. The paper also considers the results of using these glasses as shielding materials for protection against gamma radiation and the effect of doping CNDs on the strength properties and radiation damage resistance.
|
|
4- Improve the introduction by adding some recent references.
|
The authors thank the reviewer for this remark, corrections have been made to the text of the article, literature references have been added.
|
|
5- It is necessary to improve the resolution of the figures.
|
The authors thank the reviewer for this remark, corrections have been made to the text of the article, new figures have been added, the quality of the previously presented has been increased. |
|
6- Mention the importance of the finding results of this work.
|
The authors thank the reviewer for this remark, corrections have been made to the text of the article.
|
|
7- Conclusion must be improved by the results obtained.
|
The authors thank the reviewer for this remark, corrections have been made to the text of the article. During the studies of various energy gamma radiation shielding, it was found that glasses doped with CNDs with a dopant concentration of 0.1 mol are the most effective in comparison with undoped glasses. At the same time, these glasses are most effective for shielding gamma rays with energies up to 1.0 MeV, which are characterized by interaction processes in the form of the photoelectric effect and the Compton effect.
|
|
8- The English level has to be raised. |
The authors thank the reviewer for this remark, corrections have been made to the text of the article.
|

Reviewer 2 Report
In this work, carbon nanodiamonds (CNDs) are added into telluride glasses as filler to enhance the strength and hardness of the material, and the relationship between concentration of CNDs, glass strength and optical properties of the composite glass was studied. Even the conclusion shows that the addition of CNDs to the telluride glasses has positive effects on their strength characteristics and increasing resistance to external influences, I do not think the conclusion is good based on the research background in the Introduction and their expected radiation proof uses. I still want to give the authors a chance to discuss and response to my comments.
1. If authors want to use CNDs doped tellurite glasses to screen ionizing radiations, the radiation resistant property of the glasses should be investigated first, which is more important than to study the influence of adding CNDs on the mechanical property. Now, lead glasses are most popular used as red-hard optical glasses for various kinds of radiation-proof applications, including in the medical, space and nuclear fields. Comparatively, tellurite glasses, even they are heavy metal oxide glasses, have their largest disadvantage of high cost and small size than lead glasses and other silicate ones, which might throw large difficulty for their future practical use. Even like this, authors should compare tellurite glasses with other anti-radiation glasses to show why tellurite glasses should be chosen.
2. In experiment part, please give detail information how tellurite glasses are prepared and how CNDs are introduced into the glass.
“Doping of CNDs was carried out by adding CNDs to the initial mixture in a given molar ratio, after which this mixture was annealed at a temperature of 900⁰С, which is typical for the glass transition stage. “, is the initial mixture raw glass materials or glass melt? In addition, the glass transition stage is very easy to be related to glass transition temperature. Why annealed is used, rather than melted? Annealing and melting is two different thermal process for glass preparation.
3. Nano-scale fillers will be seriously agglomerated in the matrix material, therefore the dispersing state of the fillers in glasses matrix is a major factor dominating the strength and hardness properties of composite glass materials, but also will lead to decrease of the optical transmittance as shown in Fig.2. Is it possible to provide fracture surface microscopic morphology of the glass to show the distribution of CNDs. Relative to the TEM image of the presence of embedded CNDs in an amorphous glass volume showing only one diamond particle, I think the zoom-out TEM image showing more embedded CNDs are also needed.
4. Very clear in Fig.2, the increase in CNDs concentration have a negative significant effect on the optical properties of glasses. The 0.05 mol and other three intensive doping samples are no doubt impossible to be used as “optical” glasses for radiation protect. If so, there is no need to characterize their optical bandgap further.
5. In Table 2, which wavelength does the refractive index correspond to, 587.6 nm or else? And if glass samples are not polished properly, I think the band gap, Eg might be a different value, the reflection and transmission might also change.
6. Authors state that One of the important characteristics for protective materials is their resistance to long-term mechanical stress during operation, please clarify what severe operating conditions have specific requirements on resistance to long-term mechanical stress.
7. The manuscript did show the addition of CNDs to tellurite glasses has a positive effect on the strength characteristics and improving glass hardness greatly. However, their hardness is still much lower than that of silicate-based glasses and fused silica. If glass hardness and the preservation of strength characteristics are taken into focus consideration for use in author-stated severe operating conditions, I would rather suggest consider fused silica and lead silicate glass for replacement, which have good performance including red-hard, high strength and hardness, as well as lost cost.
8. The radiation resistance of the composite glass material should be provided in the manuscript, and the relationship between the radiation resistance of telluride glasses and the content of CNDs should be studied. Possibly, authors can find other applications for their CNDs doped tellurite glasses.
9. The references are totally mess.
1) No need to use ref. 21 and 22 in line 60 when authors want to tell readers what they want to do in this manuscript.
2) List all author names or use et al after the third author’s name, but not after the first one.
3) Use comma between issue and volume number.
4) Subscript and superscript are not properly used in references.
5) Long transverse line and short transverse line mixed up in glass composition.
Author Response
|
Review Report Form 2 |
|
|
1. If authors want to use CNDs doped tellurite glasses to screen ionizing radiations, the radiation resistant property of the glasses should be investigated first, which is more important than to study the influence of adding CNDs on the mechanical property. Now, lead glasses are most popular used as red-hard optical glasses for various kinds of radiation-proof applications, including in the medical, space and nuclear fields. Comparatively, tellurite glasses, even they are heavy metal oxide glasses, have their largest disadvantage of high cost and small size than lead glasses and other silicate ones, which might throw large difficulty for their future practical use. Even like this, authors should compare tellurite glasses with other anti-radiation glasses to show why tellurite glasses should be chosen. |
The authors thank the reviewer for this remark. Results of the assessment of shielding characteristics, as well as a comparative analysis with other types of glasses used for shielding, have been added to the text of the article. |
|
2. In experiment part, please give detail information how tellurite glasses are prepared and how CNDs are introduced into the glass.
“Doping of CNDs was carried out by adding CNDs to the initial mixture in a given molar ratio, after which this mixture was annealed at a temperature of 900⁰С, which is typical for the glass transition stage. “, is the initial mixture raw glass materials or glass melt? In addition, the glass transition stage is very easy to be related to glass transition temperature. Why annealed is used, rather than melted? Annealing and melting is two different thermal process for glass preparation. |
The authors thank the reviewer for this remark.
The following changes have been made to the description of the acquisition method: The addition of CNDs was carried out at the stage of sample mixing during mechanochemical grinding of the initial components to obtain a uniform distribution of elements. After stirring, the resulting mixtures were placed in a muffle furnace and heated to a temperature of 900°C, at which the mixture straightens and vitrifies when the samples are quickly removed from the muffle furnace. |
|
3. Nano-scale fillers will be seriously agglomerated in the matrix material, therefore the dispersing state of the fillers in glasses matrix is a major factor dominating the strength and hardness properties of composite glass materials, but also will lead to decrease of the optical transmittance as shown in Fig.2. Is it possible to provide fracture surface microscopic morphology of the glass to show the distribution of CNDs. Relative to the TEM image of the presence of embedded CNDs in an amorphous glass volume showing only one diamond particle, I think the zoom-out TEM image showing more embedded CNDs are also needed. |
Figure 5b shows a TEM image of a side cleavage of glasses, which shows the absence of large agglomerates of CNDs and a close to isotropic distribution of CNDs throughout the glass volume. |
|
4. Very clear in Fig.2, the increase in CNDs concentration have a negative significant effect on the optical properties of glasses. The 0.05 mol and other three intensive doping samples are no doubt impossible to be used as “optical” glasses for radiation protect. If so, there is no need to characterize their optical bandgap further. |
The authors thank the reviewer for this remark and generally agree with him on the fact that samples with a high dopant content have low optical transmission, which makes it difficult to use these glasses as optically transparent glasses for shielding. However, using them as shielding materials in places where direct visual observation or high optical transparency is not needed, these glasses can be used as shielding protective materials. |
|
5. In Table 2, which wavelength does the refractive index correspond to, 587.6 nm or else? And if glass samples are not polished properly, I think the band gap, Eg might be a different value, the reflection and transmission might also change. |
The authors thank the reviewer for this remark. To measure the optical properties, in particular the linear refractive index, the samples were polished. The determination of the refractive index was carried out at a wavelength of 587.6 nm. |
|
6. Authors state that One of the important characteristics for protective materials is their resistance to long-term mechanical stress during operation, please clarify what severe operating conditions have specific requirements on resistance to long-term mechanical stress. |
The authors thank the reviewer for this remark. The resistance of glasses to long-term mechanical stress, including mechanical pressure, impact or friction, arising during their operation as protective materials is a significant parameter. Thus, one of the requirements for materials used as shielding materials is to maintain strength and crack resistance for a long time under external mechanical pressures, friction and impact. At the same time, maintaining resistance to external influences, including exposure to the atmosphere, temperature, pressure, impact, and corrosion is one of the important parameters for long-term operation. |
|
7. The manuscript did show the addition of CNDs to tellurite glasses has a positive effect on the strength characteristics and improving glass hardness greatly. However, their hardness is still much lower than that of silicate-based glasses and fused silica. If glass hardness and the preservation of strength characteristics are taken into focus consideration for use in author-stated severe operating conditions, I would rather suggest consider fused silica and lead silicate glass for replacement, which have good performance including red-hard, high strength and hardness, as well as lost cost. |
The authors agree with the reviewer that there is a fairly large number of other materials that have better characteristics of strength and resistance to external influences. In turn, considering this remark of the reviewer, we will further study this issue and conduct a series of experiments to obtain other types of glasses doped with carbon nanostructures. |
|
8. The radiation resistance of the composite glass material should be provided in the manuscript, and the relationship between the radiation resistance of telluride glasses and the content of CNDs should be studied. Possibly, authors can find other applications for their CNDs doped tellurite glasses. |
The authors thank the reviewer for this remark. The results of experimental work on the assessment of the radiation resistance of glasses to radiation damage during irradiation with gamma rays have been added to the text of the article. To determine the resistance of the synthesized glasses to radiation damage caused by irradiation with gamma rays, several experiments including the irradiation of glass samples with gamma rays with an energy of 1.3 MeV at specified doses of 100, 200, and 500 kGy, followed by the measurement of strength characteristics were carried out. Figure 7 shows the results of changes in the hardness of the studied glasses irradiated with gamma rays with different doses.
Figure 7. Results of changes in the hardness of samples depending on the irradiation dose.
As can be seen from the data presented, doping with CNDs leads to an increase in the radiation damage resistance of synthesized glasses, which causes a decrease in strength characteristics, which is observed for undoped glasses. At the same time, the greatest decrease is observed when the radiation dose exceeds 200 kGy, which indicates the softening of materials as a result of the radiation damage accumulation. At the same time, for samples with a dopant concentration of more than 0.05 mol, a decrease in hardness for irradiated samples is observed after irradiation with a dose of 500 kGy, while the decrease is no more than 2-3 %, which is within the measurement error. Such a small change indicates a rather high radiation damage resistance of glasses doped with CNDs upon irradiation with gamma rays.
|
|
9. The references are totally mess.
1) No need to use ref. 21 and 22 in line 60 when authors want to tell readers what they want to do in this manuscript.
2) List all author names or use et al after the third author’s name, but not after the first one.
3) Use comma between issue and volume number.
4) Subscript and superscript are not properly used in references.
5) Long transverse line and short transverse line mixed up in glass composition. |
The authors thank the reviewer for this comment; all changes, considering all the comments and corrections of the reviewer, were taken into account and included in the text of the article. |

Round 2
Reviewer 1 Report
The manuscript has undergone thorough examination, resulting in writing that is simple and unambiguous, as well as an investigation that the reader will find helpful. However, I believe that the language might need additional proofreading and editing, and the references provided are insufficient; hence, I would like to request that the authors expand upon and modernise the references that pertain to the study.
Author Response
|
Review Report Form 1 |
|
|
The manuscript has undergone thorough examination, resulting in writing that is simple and unambiguous, as well as an investigation that the reader will find helpful. However, I believe that the language might need additional proofreading and editing, and the references provided are insufficient; hence, I would like to request that the authors expand upon and modernise the references that pertain to the study. |
The authors are grateful to the referee for such a high assessment of our work; the article was substantially revised and also expanded with new literature data. |
Reviewer 2 Report
I am glad to see the authors added the shielding characteristics of CNDs doped tellurite glasses and further improved the manuscript quality.
However, first of all, their response to my second comment point about how tellurite glasses are prepared and how CNDs are introduced into the glass is still not satisfying.
Authors said, “The addition of CNDs was carried out at the stage of sample mixing during mechanochemical grinding of the initial components to obtain a uniform distribution of elements. After stirring, the resulting mixtures were placed in a muffle furnace and heated to a temperature of 900°C, at which the mixture straightens and vitrifies when the samples are quickly removed from the muffle furnace.”
After mechanochemical grinding of the initial components, what type of crucible was used to hold these raw materials for heating in a muffle furnace? In open air or special atmosphere? Because CNDs are fragile to oxide atmosphere, especially at high temperature.
Do you remove the samples quickly at 900°C from the muffle furnace directly? Samples will crack under said conditions. Or they will crack during later optical polishing due to large thermal stress in glassy samples.
Secondly, I do not think it is correct as authors suggested “the main results obtained during this study can later be applied to other types of glasses, in particular those containing lead oxide or quartz”. Is it possible to maintain the state or nature of being carbon nanodiamonds after suffering from very high preparation temperature of quartz (higher than 1800-2000℃) or some lead glass (higher than 1200-1400℃).
Authors should respond to my further two comments, then the manuscript can be considered for publishing.
Author Response
|
Review Report Form 2 |
|
|
However, first of all, their response to my second comment point about how tellurite glasses are prepared and how CNDs are introduced into the glass is still not satisfying.
Authors said, “The addition of CNDs was carried out at the stage of sample mixing during mechanochemical grinding of the initial components to obtain a uniform distribution of elements. After stirring, the resulting mixtures were placed in a muffle furnace and heated to a temperature of 900°C, at which the mixture straightens and vitrifies when the samples are quickly removed from the muffle furnace.”
After mechanochemical grinding of the initial components, what type of crucible was used to hold these raw materials for heating in a muffle furnace? In open air or special atmosphere? Because CNDs are fragile to oxide atmosphere, especially at high temperature.
Do you remove the samples quickly at 900°C from the muffle furnace directly? Samples will crack under said conditions. Or they will crack during later optical polishing due to large thermal stress in glassy samples. |
The authors thank the reviewer for these comments and hope that the answers provided will satisfy the reviewer at this stage of the review.
Samples were annealed in alundum crucibles capable of withstanding heating temperatures up to 1800℃. After extraction, the samples were placed in a box with an inert atmosphere in order to avoid oxidation processes during quenching and cooling. After extraction and subsequent cooling, the samples had sufficient strength to mechanical stress during further processing and polishing. The presence of stress stresses in the structure did not lead to their rapid cracking both during processing and further exposure.
The authors also wanted to note that in the future they will conduct a separate experiment to determine the effect of an oxygen atmosphere on the oxidation of carbon nanodiamonds during glass tempering in order to determine how much this process can have a negative effect on the strength properties of glasses. In this regard, the authors express their deep gratitude to the referee for such valuable methodological instructions, which will later make it possible to carry out a number of new experimental works. |
|
Secondly, I do not think it is correct as authors suggested “the main results obtained during this study can later be applied to other types of glasses, in particular those containing lead oxide or quartz”. Is it possible to maintain the state or nature of being carbon nanodiamonds after suffering from very high preparation temperature of quartz (higher than 1800-2000℃) or some lead glass (higher than 1200-1400℃). |
The authors thank the referee for this remark. The text of the article has been corrected by deleting the phrase “the main results obtained during this study can later be applied to other types of glasses, in particular those containing lead oxide or quartz”, which actually looks very ambitious, and the authors fully agree that that for more refractory materials, the effect and mechanisms of doping with carbon nanodiamonds, as well as their behavior with increasing sintering temperature, require separate studies, which we also plan to think about in the near future.
The novelty of the work consists in the consideration of the possibility of telluride glasses modification using carbon nanostructures to increase resistance to external influences. The paper also considers the results of using these glasses as shielding materials for protection against gamma radiation and the effect of doping CNDs on the strength properties and radiation damage resistance.
|
